# “Providing Sex Education Is Challenging”: Malay Mothers’ Experience in Implementing Sex Education to Their Children with Intellectual Disabilities

**DOI:** 10.3390/ijerph19127249

**Published:** 2022-06-13

**Authors:** Nawal Nabilah Kamaludin, Rosediani Muhamad, Zainab Mat Yudin, Rosnani Zakaria

**Affiliations:** 1Department of Family Medicine, School of Medical Sciences, Universiti Sains Malaysia, Kubang Kerian 16150, Kelantan, Malaysia; nawal_kamaludin@student.usm.my (N.N.K.); rosnani@usm.my (R.Z.); 2School of Dental Sciences, Universiti Sains Malaysia, Kubang Kerian 16150, Kelantan, Malaysia; drzainab@usm.my

**Keywords:** intellectual disability, sex education, sexuality

## Abstract

Sex education (SE) is still a controversial and sensitive topic in Malaysia. Thus, individuals with intellectual disability receive less information about SE formally as Malaysian culture believes that students with disabilities should be shielded from everything connected to sexuality owing to their paucity of maturity and intellectual comprehension of the topic. Due to that reason, non-formal education by the parent is very important to fill that needful gap. This study aimed to explore the experience of mothers on the meanings of sexuality, and implementing sex education for their children with ID. We adopted a phenomenological framework and social cognitive theory to increase understanding of the phenomenon and conducted it using in-depth interviews. Twenty mothers with children aged 10–19 who were diagnosed with mild to moderate ID and/or other cognitive disabilities were involved. Almost all mothers acknowledged the influence of socio-cultural factors and religion on meanings of sexuality and practicing SE. They realized their children’s emerging sexual behavior as they go through the adolescence period. However, how they confronted sexuality and sex education with their vulnerable children was different. Three themes emerged: (1) Sexuality as an erotic experience in life, which enlightens mothers’ views on sexuality, relying mainly on sexual intimacy and sexual being; (2) Perceived their children’s sexual maturity is different, which reflects the mothers’ attitude toward their children’s sexual development; (3) Caution/hesitance in providing sexuality education, which summarizes mothers’ ways of delivering SE, either following physicality or through sociocultural ways. Our findings exemplify the need for strategies to increase parents’ understanding and awareness regarding sexuality and appropriate delivery of SE. To ensure its success, it should be a shared responsibility among all relevant stakeholders.

## 1. Introduction

Sexual health is an important aspect of a growing adolescent [1]. A healthy sexual life can contribute to a healthy body, peace of mind, and positive and respectful relationships throughout one’s life [2]. The World Health Organization defines sexual health as a state of mental, physical, and social well-being relative to sexuality. It requires a respectful and positive approach to sexuality and sexual relationships, as well as the possibility of having pleasurable and non-violent sexual experiences, free of coercion and judgment. Sexuality is conveyed and expressed in many ways such as in fantasies, beliefs, thoughts, attitudes, relationships, behavior, practice, and gendered roles [3]. Sexuality encompasses the dimensional growth of a person’s physical, mental, and social behavior. Therefore, all children have the right to be adequately informed about sex and sexuality [4].

All children and adolescents with intellectual disabilities (ID) should receive appropriate sex education (SE) suited to their comprehension levels [5,6]. Malaysia is a society that is governed by traditions in which children accumulate significant knowledge in their life through their everyday experience from their environments and social and cultural norms. They have a limited perspective on the gist of sexuality and SE owing to cultural customs and a limited conceptual understanding of religious beliefs. Sexuality is considered any behavior connected to sexual intercourse [7,8]. Therefore, the majority of Malaysian parents do not discuss sexuality or sex-related or body functions with their children in an open and frank discussion. An earlier study involving adolescents with disabilities revealed that their parents did not openly discuss sexuality and they preferred sharing concerns with a colleague [9]. Similarly, research indicate that most Malaysian parents reared in culturally conservative surroundings have a negative view of their children’s sexual awareness [10].

In Malaysia, SE is informally taught to Malay children in terms of appropriate personality traits and desirable behavior—to behave in ”feminine” ways for girls and “masculine” ways for boys around four years of age [11]. They are also taught that there are topics that cannot be talked about, including sexual intercourse, private body parts, and sexual bodily functions [12]. Islamic teachers teach SE when the children approach teenage years, mainly on menstruation issues, dress code, and avoiding immoral activities or illegal conduct topics according to the law of the country and social norms of the community. They have also been taught about signs of *akil baligh* for girls and for boys. It is important for them to know about these signs. *Akil baligh* is a term used in Islamic legal terminology to indicate someone who has achieved puberty and is fully responsible under Islamic law. When the person approaches the end of their teenage years or prior to marriage, the roles and rights of a married couple are taught in formal *munakahat* classes or workshop classes (pre-marriage courses) [13]. This is made possible by the law that requires all Muslims applying for marriage in Malaysia to have a certificate of having completed this course [14].

Formal SE is not easily implemented in the Malaysian school curriculum. The government started to integrate SE into the national curriculum for secondary schools in 1989, while for primary schools it was started five years later. The SE module was integrated into various academic subjects such as in human science, Islamic education, languages, and moral education [15]. However, the implementation creates debates among society up to the scholars’ level, partly due to the incomplete training of the teachers who are supposed to handle the topics [15]. After taking into consideration the sensitivity of the topic in the community, the SE module has been renamed to become Reproductive Health and Social Education) or *Pendidikan Kesihatan Reproduktif dan Sosial* (PEERS) in 2011 and delivered as a part of the Health Education subject [15].

Despite these efforts, reported misconduct among school children with ID persist. Lack of understanding among the parents about the implementation of SE at school was found to be one of the reasons causing the low effectiveness of the program to educate special needs children in this matter [15]. SE for students with learning disabilities has not been implemented formally [16]. SE is still a controversial and sensitive topic [16]; thus, individuals with ID receive less information about SE as Malaysian culture believes that students with disabilities should be shielded from everything connected to sexuality owing to their paucity of maturity and intellectual comprehension of the topic [16].

Sexuality issues of persons with ID are still misjudged by society even though this group exhibits comparable sexual desires akin to the general population [16]. These urges drive them to be attracted to sexual partners and engage in romantic relationships and sexual activities [6]. Some parents are unwilling to view their children with ID as sexual beings. A study showed that parents with children who have ID are less likely to perceive their children engaging in sexual acts during their adolescence. However, they still believe that providing SE to their children is a must [17]. In another study, all mothers ignored SE for their children due to the belief that their children with ID should not have a sex life [18].

Globally, epidemiological research shows that ID affects about 1% of the population [19]. As of 2020, there are about 20,300 new cases of children with disability registered with the Department of Social Welfare in Malaysia [20]. However, there are no accessible data on the prevalence of children with ID in Malaysia [20]. Malaysia has 28.2% of children under the age of 18 in 2020 or 9,247,500 children out of a population of 32.7 million. Male children (4,774,900) outnumber female children (4,472,600). Kelantan has the third largest proportion of children under the age of 18 at 35.2% [20].

Intellectual disability is characterized by deficits in mental capabilities that impede adaptive functioning in three domains: social, conceptual, and practical. These deficits emerge throughout the developmental stage [21]. Several neurodevelopmental disorders or medical diseases coexist with ID in individuals, including autism spectrum disorder (ASD), attention deficit hyperactivity disorder (ADHD), or disorders such as cerebral palsy (CP) [22]. ASD, which has a comparable prevalence to ID, is defined by a deficit in social interaction and communication, restricted interests, and repetitive behaviors. At least 25% of children with ASD also have ID [23]. Genetic factors also play a significant role in ID. The most common chromosomal abnormality associated with an ID is Down syndrome.

There are a lot of health concerns that arise due to limited knowledge of SE. Children with ID are vulnerable to sexual abuse. They suffer from sexual abuse more often than their able-bodied and able-minded peers; furthermore, most cases of abuse happen repetitively within the close social environment of the victims [24]. By providing SE to them, they will be aware of situations that threaten their safety. Exposing them to appropriately planned sex education at an early age can help them live in secure and safe environments [5].

Parents play a pivotal role in providing children with SE; children rely entirely on their parents for information from an early age. It is necessary and fundamental to form a partnership with parents and develop mutual understanding and responsibility for children’s success in the educational system [15]. A review of the relevant literature, particularly in Malaysia, reveals that this topic is underexplored. Very little is known about parental understanding regarding sexuality and interaction elements that influence their behavior in providing SE to their children with ID. The reviewed study focused mostly on the perspectives of educators or on parents of able-bodied adolescents [25]. Thus, this qualitative study aimed to explore mothers’ understanding and experiences related to sexuality issues among their children with ID. It is essential to explore parental engagement in the delivery of SE to empirically identify parental knowledge, abilities, and the challenges related to SE to plan for future intervention.

Based on the concept that approaches to SE and sex education can be learned, regulated, and motivated, we chose social cognitive theory (SCT) as a base of our research framework and guidance for method and data analysis, presented our findings, and provided direction for future research [26]. Human behavior can affect or be affected by both environmental and personal factors. All these three factors are interrelated as shown in Figure 1. The SCT identifies three major interacting factors that have a bidirectional impact on mothers’ outcomes: personal, environmental, and behavioral [26]. This two-way interaction is shown in the ‘reciprocal triad interaction model’, where (1) people can create their own environment—what sexuality meant to them, how important SE is to their children, and their self-belief on this issue, (2) people can be affected by external environmental stimuli, such as the way they perceived Islamic teaching and *Adat* related to the sexuality, the ability of their child to accept SE, as well as supports and barriers they may experience, and (3) people can also make choices about whether or not to proceed with actions, such as educating their ID children on this topic [27].

The strength of any influence differs between people and events. The three elements of the reciprocal triad include: the environmental factors, which include culture, religion, politics, laws, etc.; the personal factors, which include cognition such as self-efficacy and outcome expectancy, emotions, and physical appearance; and behaviors, which include actions, verbalizations, and decisions. This theory is particularly helpful to comprehend major significant interactional relationships between the different factors that influence mothers’ perceptions of their children’s sexuality and behavior about providing SE, which is the aim of our research.

## 2. Materials and Methods

This study was conducted using a qualitative method wherein in-depth interviews with a phenomenological approach were used to explore mothers’ views on sexuality and sex education for their children with ID. The qualitative method enables the researcher to study selected issues comprehensively without being constrained by pre-determined categories of analysis and allows the participants to voice their opinions [28].

The phenomenological approach allows the researcher to delve into the perceptions, perspectives, understandings, and feelings of mothers who have experienced the sexuality phenomenon with their children with ID by describing “what” and “how” they experienced it [29]. Parents of children with ID at age 10–19 years (early/middle/late adolescence) and able to communicate well in Malay were included in this study.

### 2.1. Settings

This study was carried out in Kota Bharu, Kelantan, a sub-urban state in Peninsular Malaysia’s northeast. The Kelantanese Malay people are the state’s majority ethnic group, accounting for 94% of the state’s population, and are Muslim [30].

### 2.2. Participants

Mothers of adolescents who had mild to moderate ID or other neurodevelopmental disorder with comorbid ID (ASD, ADHD, DS and mixed disorders of ID) were enrolled in this study. They were recruited via word of mouth from relevant NGO leaders, and by utilizing snowball strategy with Child Psychiatry Clinic, Hospital Raja Perempuan Zainab II, Kota Bharu, Community-Based Rehabilitation (CBR), and any related organization. A key person (a medical officer, teacher, or a manager) distributed the flyers and invitation letters to the parents personally or via social media group.

Those who were interested in participating directly contacted and provided their phone numbers to the key person so as to be contacted by the primary researcher. Once mothers agreed, the date and time of the interview were set up according to the participants’ schedules. The process of recruitment stopped after the saturation point was reached where there was no newer data emerged. Data saturation for this study occurred after the 18th interview and we allowed another two interviews to be held for confirmation.

### 2.3. Procedure

This study used face-to-face and in-depth interview techniques [19]. The interviewer was a female Family Medicine Trainee with no prior relationship to the participants [22]. On the day of the interview, consent was obtained from participants before the interview. Then, participants were asked to fill in the sociodemographic form. The interview session began once they completed the form using a semi-structured questionnaire. This study was conducted in the participants’ native language (Kelantanese dialect) to enable participants to share their views and experiences comfortably and allow the researcher to grasp subtle and concealed nuances in the participants’ utterances.

The questions were open-ended, clear, and understandable, conforming to SCT (Table 1). We began by asking the participants the following key questions: “From your experience, what does sexuality mean?”; “What is your view about sexuality in your children?” and “How do you provide sex education to your children?” We also questioned how they saw their children as sexual beings, the gendered role as assigned, and how they conveyed SE to their child with ID as compared with their able-bodied siblings. These were then followed by more specific questions.

The time frame for the interviews was flexible and most usually took around 60 min. The interview took place at a convenient, quiet location that was free from distraction, either at a hospital during follow-up, or at the children’s community-based rehab center, or anywhere they preferred. All the interviews were audio-recorded, stored in a safe place, transcribed into text, and analyzed later. A pilot study among three mothers was conducted with senior co-researchers who were experts in the qualitative study prior to the actual study to test the validity, acceptability, and suitability of the questionnaire.

### 2.4. Data Analysis

The interviews, which were audiotaped, were transcribed verbatim. Then they were entered in NVivo (Qualitative Research Computer Analysis Package) software for coding. Thematic analysis was used to extract pertinent themes in this study from the transcribed text. To begin, the three researchers (N.N.K., R.M. and Z.M.) reread the first five transcripts attentively to become acquainted with the mothers’ overall ideas and viewpoints. Following that, the primary researcher generated an initial list of codes in NVivo^®^, and we coded the transcripts in a meaningful and systematic way.

We first looked at each transcript individually before grouping the discovered themes to create an overall group analysis and arranging them into an interrelated structure (themes, subthemes, and axial coding). To ensure validity, reliability, and coding correctness, the research supervisors (R.M. and Z.M.) checked all interview transcripts’ codes with the primary researcher (N.N.K.). Any conceptual disagreements about thematic analysis were reviewed and modified, and other preliminary themes were created.

Additionally, a co-researcher (R.Z.) with interest in children’s sexuality reviewed and discussed the coding and preliminary themes for all transcripts together to provide general constructive comments. Finally, the study team achieved agreement on the final codes, as well as on the themes, subthemes, and axial coding. The transcribed text was emailed to be read by selected participants to validate the findings. Any different possible interpretations were discussed. None of them disagreed with these findings.

## 3. Results

### 3.1. Characteristics of the Participants

Twenty-three eligible mothers were contacted by phone; however, three were unable to participate due to scheduling difficulties. Twenty Malay mothers of adolescents with ID took part in this research. Table 2 describes the characteristics of the mothers involved. The mean age of the mothers was 48.8 years (SD, 8.45). The majority of them were married (70%); more than half (55%) had a secondary level of education and were homemakers.

The mean age of the children was 15.4 years (SD, 2.62) with an equally distributed representative from young (35%), middle (35%), and late (30%) adolescents. The majority of them (75%) were male, with half of the adolescents having ID and DS, one-fifth (20%) had ASD, and one-third of them (30%) had mixed diagnoses. Most mothers had three other able-bodied children.

### 3.2. Themes

Three primary themes emerged from the mothers’ experiences: sexuality as erotic experience and life, perceived their children’s sexual maturity as different, and caution in providing sex education. SCT and phenomenology explained the experience of these mothers on their meanings of sexuality and understanding of SE. The analysis showed the influence of cognitive and behavioral factors on how mothers provide SE to their child with ID. This interaction can be clearly seen through the themes below.

#### 3.2.1. Theme 1: Sexuality as Erotic Experience in Life

When we asked what sexuality means to them, the researcher discovered that all 20 mothers had varying perspectives about sexuality. The individuals’ interpretations of sexuality were influenced by personal understanding, sociocultural beliefs, and socio-cultural environmental influences. Some mothers understood sexuality in terms of personal relationships and its connection to marriage, while others took a more holistic approach. The culture and the religion of Malays show how this construct can mold the experience of sexuality in Malay mothers as many claimed sexuality is confined to intimate relationships.

Sexuality is an intimate relationship

Many mothers were just not sure of what sexuality is. On further questioning, some regarded it as something that might be akin to sexual intimacy, while others asserted that it is about marriage.

Nine mothers equated sexuality with sexual intimacy. P10 said sexuality encompassed sexually intimate relationships, which then were fueled with sexual desire and resulted in sexual intercourse. Even she had previously read about sexuality on social media or seen it on television; sexuality, to her, refers to sexual intercourse.

“*Sexuality is… how to say… it is about the sex of that person… those teenagers have some desire which is more to sexual need itself.*”(Mother of a 19-year-old male with mild ID, homemaker)

Despite being unsure of the meanings, three mothers described sexuality as related to the inclination to become married and legitimize what is generally forbidden in religion. According to P11, when men and women share a particular feeling, the relationship’s overall purpose is marriage. Once they are married, men and women are free to do anything they desire as the partnership is deemed halal in Islam. In Islam, halal refers to what is permissible under Islamic law.

“*It’s probably related to the marriage wishes. It might be something that has to do between men and women. It refers to their personal needs which also want to get married and so on.*”(Mother of an 18-year-old male with mild ID, homemaker)

Similarly, P17 viewed sexuality as an emotional bond between two people that will eventually lead to marriage. During the interview, she referred to the phrase “find another life partner” as “marry.” Following their marriage, they are permitted to engage in sexual activity and sexual intercourse.

“*It means the feelings as much as people said… those are our emotions that are connected to each other. Err, it is more to find another life partner. It’s like that. Yes, it also means the relationship will go further… it will lead to intimacy and sexual interaction.*”(Mother of an 18-year-old female with mild ID, traditional massager)

The majority of mothers who have a little understanding of sexuality are mostly homemakers with a low-to-moderate level of education. While some of them s knowledge on sexuality via reading, watching television, or surfing social media, they could not adequately define the term.

Sexuality is a part of sexual being

While most mothers defined sexuality as intimacy, six educated mothers, particularly those working in the medical line, described sexuality beyond sexual intercourse and heterosexual relationships. Sexuality encompassed all aspects of life, which include sexual development, sexual orientation, gender identity, physical and emotional intimacy, and reproduction. P2 shared her views below.

“*Sexuality means… about body organs related to reproduction. Err…which part you can touch… and which part you can’t. How the process happens inside like biological, physiological processes. It is not about sex. What’s the difference between a man and woman, it also indicates the process of puberty… if a woman, it is about the menstrual cycle and more… the pregnancy processes. What will change when they become teenagers.*”(Mother of 12-year-old male with ASD and ADHD, lecturer)

Another mother, P9 expanded the meaning of sexuality beyond physicality. It includes emotions and the effect of sexual behaviors among teenagers.

“*It is also about our biological sex-anatomical parts of our bodies, gender identities, either you are a boy or girl, and the boundaries or limitations of association between girls and boys, the ability to have any emotional expression, what is sex, and the consequences of sexual activity. Sexuality encompasses all aspects of life.*”(Mother of a 16-year-old male with ASD, teacher)

Uncertain meanings

However, two mothers did not comprehend what sexuality is or were still confused about the concept of sexuality. They asserted that they were unsure of the concept of sexuality due to a lack of exposure to information regarding the same in the past and a lack of conversation about sexuality among family and friends.

P5, among the two mothers, overheard friends’ conversation by chance regarding sexuality but never asked or interrupted to find out the details. She is a stay-at-home mother with a low educational background. Her life revolves solely around her children and spouse. Even though the topic was less sensitive to her, she made no attempt to discuss it with friends or family.

“*I don’t understand. I’ve heard from my friends. But I don’t have any idea what’s it all about.*”(Mother of a 15-year-old female with moderate ID, homemaker)

#### 3.2.2. Theme 2: Perceived Their Children’s Sexual Maturity as Different

Mothers perceived their children’s sexuality through a variety of lenses according to what they have gone through in the past, their educational levels, and how they understand sexuality from culture and religion perspectives. Nineteen mothers stated that their children’s sexuality is emerging but from a different perspective. Sexual attraction, viewing of pornographic material, self-stimulation, and expressing a desire to have sex are all examples of sexual behaviors displayed by their children, which sometimes go unnoticed by the parents. Mothers of boys reported observing morning erections and nocturnal emissions. However, they felt that the development of sexual behaviors is delayed compared with their able-bodied children, while a few were unaware of these changes.

Unrecognized children’s sexual maturity

A few mothers did not anticipate their children to show sexual behaviors. They were unaware that their children had previously experienced sexual development as similar to other siblings. Only when the researcher raised questions or prompted the mothers did they recall an event that happened in their children’s life as they lack the focus and failed to observe their children’s sexual development since this is something which is not commonly done in the community, especially in children with ID. For example, P2 realized her child’s sexual desire when he saw a picture that depicted kissing.

“*When he looked at the kissing photo, he showed some responses. That means he must know something, right?”*(Mother of a 12-year-old male with ASD and ADHD, lecturer)

In fact, one day, her son informed her that he had accidentally seen a nude girl online. Someone on YouTube suggested he looked up nudist. While playing an online game, her son became acquainted with him. She was concerned that this move might have enhanced her son’s sexual desires. The mother was then asked about her son’s responses after seeing the video.

“*He said that the girl was naked, but he also told me he could not see her much. I do not really understand it. I did not look at his response when he saw that because he did tell me after he was done. I thought that he said it to me because he wanted me to know, and he then closed that channel.*”

The mother believed that he had no intention of seeing that naked woman. Until that moment, the mother did not acknowledge the sexual urges that were present in her child. She confessed her son’s naivety. He would never engage in a behavior deemed inappropriate by society. Pornography and nakedness are regarded as immoral in Malay culture and haram in the Islamic faith. The mother is concerned that her son’s sexual urges would break religious laws and be deemed unacceptable in cultural beliefs.

Despite the denial of her son’s sexual maturation and behavioral changes that occurred during adolescence, another mother, P3, recalled one event that occurred during her son’s puberty. This improper sexual behavior at school was brought to the teacher’s attention.

“*The teacher called me and said… My son wanted to go to the toilet. The toilet was on the other side. So, he went there. A girl was already inside the male toilet. He said that he wanted to pee… but that girl refused to go out. He tried to open his pants because he could not hold it anymore, but the teacher saw him at that moment. Then, the teacher claimed that he wanted to touch that girl.*”(Mother of a 15-year-old male with mild CP and ID, homemaker)

When she asked her son to clarify the accusation, the son confessed to her that he behaved like that because he was experiencing arousal at the moment. The mother then realized that her son, like other able-bodied adolescents, displayed sexual urges at this age.

Similarly, mother P15 saw her daughter’s habit of placing something between her private parts while lying down. Initially, she believed the habit was for her comfort, but her mother subsequently surmised that her daughter was indulging in such behavior to satiate lustful urges. This habit emerged as the girl reached puberty. In the mother’s eyes, sexually explicit occurrences were often deemed unsuitable or unpleasant in children with disabilities. As a consequence, it is possible that when their children display changes in sexual behavior, the parents do not notice.

Expecting delay in children’s sexual maturity

The association that the mothers established between environmental factors in viewing their children’s sexuality was reinforced when asking about their children with ID’s sexual development. Some mothers expressed that their children’s sexuality might be delayed in comparison with able-bodied teenage development, especially in non-physical sexual maturity. Emotional sexual development, such as engaging in a romantic and sexual relationship or falling in love, is one of the examples of their children’s non-physical maturity. Seven mothers out of twenty reported that their children had an emotional attraction to the other gender. The majority of children showed their interest in the other gender between the ages of 11 and 12. In contrast, P1 confessed that her son had never shown interest in girls at this age.

“*However, he did not have interest to become friends with any girl (laughing)… he didn’t feel it. His social life is quite challenging. So, from his interactions among girls and boys, he would prefer to be closer to the boys as he never mentioned any of his girlfriends. Never heard of it.*”(Mother of 12-year-old male with ASD)

Similarly, P2 said that her son did not tell her about a girl he liked at school. She said that when she teased him about whether he likes or dislikes that girl, her son clearly did not understand it.

“*In terms of feeling, I never heard him mention his feelings toward other girls… Never heard of it.*”(Mother of a 12-year-old male with ASD and ADHD)

On the other hand, a few mothers were more concerned about the signs of puberty in their children. For example, mother P15, out of her curiosity, took an action to confirm the maturity status of her son. She regularly checks her son’s mattress in the morning to see whether or not he has attained akil baligh (puberty).

“*Just because he got circumcised at the age of 12, and now he’s 19, I’ve never found any semen on the mattress… I have to observe frequently as an Islamic teacher told me that he is very special because he does not have a desire for sex. I felt so relieved. Until today I always ask my other children whether Aiman has reached puberty or not. In terms of his voice, it is already changed but in terms of a wet dream, there’s no sign yet.*” (Mother of a 19-year-old male with ASD, homemaker)

One week following the interview, the mother contacted the researcher to inform that she found semen on her child’s bedding that morning. She was overjoyed since her son, in terms of maturity growth, is comparable to an able-bodied adolescent even though slightly delayed. Wet dreaming or nocturnal emissions were long-awaited in the boy and one of akil baligh’s signs. Similarly, P19 was anticipating a similar event in her son’s life after knowing from the doctor that her son would attain puberty at his current age. However, it has not happened yet.

However, one mother in this study is adamantly opposed to seeing her child as a sexual being. She felt her child was asexual even though he was entering adolescence as she understood from a socio-cultural point of view. P10 perceived her son as asexual, meaning he has no sexual attraction to people of any gender and no desire for intimacy with another person and is not showing any sexual attitude despite his age of 19. She said, *“So far, I do not see any changes in terms of sexuality. I don’t think so”* (Mother of a 19-year-old male with mild ID, homemaker).

Children with intellectual disabilities carry social emotions

While claiming delays in children’s sexual maturity, all mothers noticed their children imitate the social emotions related to sexuality much earlier. It was observed that the children felt embarrassed if they saw a kissing scene on television, ashamed when they came out of the toilet nude and were seen by other siblings, and uncomfortable when their mother playfully touched their bodily parts. Malay children are taught to be appropriately ‘malu’ (shy, humble, and modest), which is viewed as highly moral and demonstrating good social emotions according to culture and religion.

P6 said that her daughter would automatically shut her eyes if she saw an indecent scene on television and she was aware that displaying such sexual actions was not an appropriate thing to watch. P2 said that her son would cover his body right after showering to avoid being seen naked by his siblings as it was shameful. According to the mother, as a Muslim, awrah must be kept hidden and it has social boundaries that must be respected, even within his own family.

“*He said that his little brother/sister might see his private part. So, when he wants to take a bath, he will make sure that he wears his towel, although sometimes I know he does not wear it too when his brother/sister is not around.*”(Mother of a 12-year-old male with ASD and ADHD, lecturer)

P14 claimed that her daughter was apprehensive to physical touch i.e., tickling when she was 12 years old. She believed that it was a good sign that her daughter had an internal sense of shame. This realization is crucial to avert the possibility of incest in the future.

“*After she reaches puberty, we cannot touch any of her body parts. She said it’s a sin if we touch the private part of others… it’s not good for us (to do so). So, she turns out to be angry because she feels unsecured.*”(Mother of an 18-year-old female with ID, cleaning supervisor)

Apart from practicing good social emotions, all mothers stated that their children do not have any identity crises. Self-recognition of gender identity in their children is important to promote and encourage appropriate social development.

#### 3.2.3. Theme 3: Caution/Hesitance in Providing Sexual Education

Almost all mothers felt that early sex education is important for children with ID, and it should be continuously reiterated until adulthood. Mothers shared three different ways they preferred to deliver sex education, and they are as follows: it should be appropriate for the children’s age and development, the societal and cultural norms, and mothers’ convenience.

Focus on age-appropriate topics

Sex education covers a broad range of aspects that are tailored to the child’s age and developmental stage. All mothers agreed that safe body touch, friendship, and gender identity and role are a few topics that a preschool-aged child needs to learn about. These topics are applicable for older children as well, although they are taught at a more advanced level.

The main lesson they were usually taught at home is safe body touch. P8 also taught it herself but chose to wait until her son reached adolescence.

“*I told him that if someone tries to do something bad to you … touch your butt, you cannot befriend that person anymore. You must try to avoid them and never stay there with them anymore. I also told him to go away if somebody tries to do inappropriate things to him in the toilet. Please don’t get involved with these kinds of people.*”(Mother of a 17-year-old male with ID, nurse)

Conversely, P4 taught her son about appropriate touching from an early age, even with their siblings. *“Then, if you play with your sister, you cannot touch her breast or her vagina or her butt. You cannot touch her body parts like that.”* (Mother of a 12-year-old male with ASD, lecturer).

Care for breasts and pubic hair are topic of concern for female children during the early puberty period and then followed by care during menstruation. P15′s daughter started menstruating at 9 years old. When she had menarche, her mother began to teach her the appropriate way of handling pads and taking care of her private parts and breasts. It is not an easy task at first, and she just taught her daughter repeatedly until she could perform it independently.

“*She did ask me why she had some hair under her private parts. It is because she is getting close to puberty. She has to shave them at least once a month. She cannot just let them be because her body will smell bad. If she gets showered, she must wipe her body properly. I bought her a set of disposable shavers. Her breast buds are developed too. So (I told her) she has to wear a brassiere.*” (Mother of a 15-year-old female with DS, cleaner)

When asked about their teaching activities on other reproductive issues such as the sexual relationship between different genders, all mothers felt it was inappropriate to teach during adolescence. P1 shared her fear that such a conversation might encourage her son to find more information on the issues and dwell in sexual activity. She would instead wait for her son’s inquiry on that subject.

“*It is not his interest. He did not ask. If we told him about that … for example, sexual thingy … what is sex? … Penetration, penis, vagina, etc. I usually avoid telling him about them. I try to avoid it because it is not their level yet to learn about it. If we go into details, he just can’t. If he gets obsessed with it, everything will be finished.*”(Mother of a 12-year-old male with ASD, lecturer)

Similarly, none of them touched upon the topic of contraception because they felt it was too early to be discussed.

Prefer figurative, replacing taboo words

In Malay culture, words describing sex organs and sex are highly taboo. The society often replaces *zakar* (penis) as *pelir, kote, anu, awang, batang, punai, burung, konek*, and *kotek*. This could be clearly seen among mothers in this study. Many mothers used figurative language to refer to the reproductive organs instead of using the scientific words such as a penis or *zakar* when talking with or educating their children, as is customary in Malay. P4 said when she explained sexual reproductive organs to her children, she used symbolic language. It was not wrong for her since Malay culture stipulates it.

“*When he was a nine-year-old, he asked to be circumcised. So, when he said that, I asked him what he knew about that. Does he know which part needs to be removed, what he should do, which foreskin will go under surgical removal. He said that he’s okay. Some of his friends told him that they will only remove some foreskin from his ‘burung’ (bird, word for penis). Then his ‘burung’ will grow larger as he will also grow up to be an adult. He told me so.*”(Mother of a 12-year-old male with ASD)

Similarly, P1 described sexual anatomy to her son using figurative terms. She has to educate her son using a lot of terms to expand his vocabulary. However, using correct terms such as zakar is not ideal since it sounds close to a birdcage in Malay and may create confusion for the child.

“*When I explained about his body parts, I called it ‘bird’ or ‘konek’. He should know many terms of it. I taught him in English and also Malay language. Not just limited to one or two words only. He cannot pronounce that word at home. We cannot pronounce the word ‘zakar’ as it sounds like a ‘sangkar’ (birdcage). He might get confused and it will be difficult for him. It’s too hard to be remembered and instructed.*”(Mother of a 12-year-old male with ASD, lecturer)

In contrast, P2 used the correct terminology to define the private parts when talking to her son. Formerly, she used euphemism terms for genitals, but after learning about sexuality in children, she began to use the correct words. She said that the advantage of using proper terminology was that it facilitated communication between parents and children.

“*I taught him to use the real words because my son used the standard Malay language or English. But if there are any of his friends who use different words, I will explain further those words themselves.*”(Mother of a 12-year-old male with ASD and ADHD, lecturer)

Use pertinent opportunity

Most mothers in this study used real-life experiences to provide sex education informally to their children. For them, teachable moments are everywhere. One mother, P2, used her birth experience to teach her older son with ID about reproductive health. She needed to use a visual medium, such as videos, to explain how the delivery process works.

“*He was 9 when I gave birth to his youngest sister. So, he just got a baby brother idea back then. He also had two elder sisters. But he had no idea that I gave birth to his half-sisters! For the simple reason, he did not realize that babies are born to mothers. After seeing me pregnant and giving birth, he knew he was now a big brother. He inquired about his birth. Because he was born by Caesarean section, I played him a video clip. Then, he got it!*”(Mother of a 12-year-old male with ASD and ADHD, lecturer)

Mothers also incorporated religious rules when providing sex education and governing children’s sexual behaviors. The words halal and haram are the usual terms used in the Quran to designate the categories of lawful or allowed and unlawful or forbidden actions, respectively. By making those connections to their religion, their children will be more submissive and obedient to what is taught. P15 said:

“*I told her that God does not allow us to show our body parts to others. If you go outside and you don’t wear a hijab, people can see your hair and your breast. It is very important information which she must know. If we do any good deed, Allah will reward us with good rewards. If we commit sins, Allah will punish us thereafter. That’s all I’ve said.*”(Mother of a 15-year-old female with DS, cleaner)

Moral development is a key aspect of early childhood education. As P13 remarked, she taught her son about gender identity and physical boundaries while engaging with those of opposite gender, which also were bound by Islamic social etiquette.

“*When he was in standard 5 [11 years old], I taught him not to play with girls. He could not touch any girls because it is prohibited in Islam. If he contacts the boys, then it is okay. Since we taught him to shake hands with others in the morning, we should also remind him not to shake hands with girls. I showed him the difference between male dresses and female dresses too.*”(Mother of a 14-year-old male with DS, homemaker)

Most mothers did not really initiate sexual discussion with their children. They started talking when their children did anything sexually inappropriate or upon asking about sexual matters. P3 utilized her time with her son while driving him to school to talk about the matter especially after she received reports from his teacher with regard to his inappropriate sexual behavior.

“*Do not try to kiss the girls. It’s not proper to do that. Their mothers must be mad when they know about it. Sometimes when I send him to school, I keep reminding him to behave well and do not disturb any girl. Never touch them or do something else. It becomes my habit to remind him almost every day so that he will understand.*”(Mother of a 15-year-old male with CP and ID, homemaker)

Another mother, P19, noticed that her son saw indecent content while browsing the Internet (either intentionally or by pop-ups that show nudity). She then used it as a springboard to talk about it spontaneously. She believed talking directly while he was still browsing is the most ideal way to advise and approach her son with special needs regarding sexual exploration on the internet and its rules.

“*After knowing that he watched some porn indirectly, I told him spontaneously because my son has his way to adapt with advice. I have to make sure I provide the proper time for him to advise on good characters or attitudes. I could not take time and sit together to talk. If I see him acting weird, I will straight away tell him what he is supposed to do. For example, if he is watching videos, I will remind him that he should not watch weird videos because it is not good for him.*”(Mother of a 14-year-old male with ASD and ADHD, homemaker)

## 4. Discussion

In this study, almost every mother acknowledged their children’s emerging sexuality as they go through adolescence. However, how they confronted sexuality in their children was different. The way mothers perceived sexuality in their children will influence the extent to which SE is provided. We found that personal understanding, sociocultural beliefs, socio-cultural environmental factors, and mothers’ practices have a significant impact on the views of mothers on sexuality and how they educate children with ID.

Social cognitive theory (SCT) provides a better understanding of this phenomenon by highlighting the effect of personal factors, socio-cultural norms, and behavioral factors on mothers’ choice to SE practices [26]. In this study, how parents perceive sexuality in their children determined how SE is implemented in their daily lives (Figure 2). Sexuality became understood as a consequence of the interrelation of three reciprocal triad components.

### 4.1. Influence of the Environmental Factors

Regardless of their awareness of their social conditioning, the mothers in this study were exposed to several environmental factors related to sexuality that promoted the idea of being a “competent mother” but “sexuality not to be stressed during childhood because of its meaning”, causing providing SE at home to their children being quite challenging. These factors included: Islam; *Adat* (social norms); child factors; and the socialization that occurs in multicultural societies related to their sexual health, either bringing support or providing a barrier for SE. The impact of the resulting struggle between such factors and their influence on the capability to provide SE to their children with ID was very clear through examining of these factors. Without appropriate knowledge and empowerment in formal SE, *Adat* and Islamic knowledge retained a strong influence on these mothers [7].

*Adat* perspective on children with ID can be observed from mothers’ views about their children’s development of sexual behavior. When it comes to viewing children with disabilities as sexual beings, society highlighted children with ID as ‘*anak syurga*’, (Children of Paradise). Children with disabilities are very close to God as many of them innocent and less capable of doing something immoral as compared with able-bodied children and if they did, it is not considered sinful. This, for many mothers, may cause them to continue to misjudge children unfavorably and stereotypically as asexual, or as perpetually aroused and unable to regulate their sexual desires [31]. Our findings are congruent with Stein et al., i.e., mothers were less likely to believe that their child will have sex during adolescence [17]. Pownall et al. showed that mothers are extra cautious in their attitudes toward sexuality [32]. According to Pownall’s study, mothers believed that their children were not prone to be engaged in romantic relationships and had fewer sexual feelings than average adolescents. As a result, mothers may have thought that providing such information is meaningless and may potentially raise anxiety in their children.

Parents’ involvement in providing SE is also related to environmental factors such as the influence of community advice or information they received in lived experience. Our study found that institution and healthcare play an important role in parents’ understandings of sexuality issues in their children. The more precise and reliable the information they receive, the more they will comprehend the sexuality problem of their children with ID. This finding is on par with other studies that a lack of knowledge about sexuality is due to a lack of community support which is due to socio-cultural beliefs and insufficient material and moral resources offered to them [33,34,35]. Parents need to be supported and empowered to foster the meaning of sexuality. Research revealed that parents of adolescents with intellectual disabilities need adequate knowledge, attitudes, and practices to successfully address sexuality concerns [36]. Therefore, it is vital to increase the visibility of children and adolescents with disabilities in the community, including their sexual needs and desires, to combat stigma and false beliefs, and increase sexuality awareness.

### 4.2. At the Personal Level: Competing Cultural Norms and Knowledge on Sexual Health

This study revealed that mothers had a limited grasp of the meaning of sexuality. Sexuality as an erotic experience and life reflect mothers’ personal factors based on their knowledge, life experience, self-belief, and cognition. As their children reached puberty, they endured a variety of physical changes, including the development of pubic and underarm hair and a rise in body odor. For females, these changes include the growth of breasts and the onset of menstruation; for males, the testes develop. However, their disproportionate cognitive capacity and physical maturity kept their parents in a state of denial. Previous literature has shown that parents who have children with ID had a poor understanding of sexuality when questioned, and this will leave parents perplexed as they encounter their children’s developing sexuality [37,38,39].

Our study also found a common misunderstanding among people that sexuality and sexual intercourse were synonymous and which was influenced mainly by environmental factors [40]. This misunderstanding is similar to that found in the study by Do et al. and studies among Malaysian women with female sexual dysfunction, where most Vietnamese parents and Malaysian women were unaware of the true meaning of sexuality [7,41]. In their mind, they considered sexuality to be limited to sexual intercourse, which creates fear to implement SE. Malaysia, like Vietnam, is an Asian nation with strong socio-cultural traditions that view sexuality as a sensitive topic to talk about, discuss, or even learn about.

Despite feeling embarrassed and fearful to discuss SE with their children, we found mothers who have some knowledge of sexual health realized the need SE at home for their children better. It is similar to what Azira et al. discovered, i.e., a statistically significant relationship between parents’ sexual knowledge and educational background on parents’ understanding of sex education [25]. Another research found substantial disparities in knowledge across groups of parents who had completed middle school, vocational school, or university [42]. In our study, we found that mothers’ understandings of sexuality were influenced by their educational backgrounds. When we asked about the meaning of sexuality, most mothers with a diploma and high educational attainment could respond adequately as compared with those with lower academic status.

### 4.3. At the Behavioral Level

At this level, we realized that the competing influences between environmental factors and personal understanding of mothers and their beliefs toward sexuality and the need for sexual education for their children lead them to the multiple ways of delivering SE at home and how deep they will share about it. Differences can be observed between these two types of mothers in this study, namely, the educated and the less educated.

#### 4.3.1. The Educated Mothers

The educated mothers, those who predominantly had higher education and better understanding of sexual health, were found to have better attitudes toward their children’s journey to adolescence in terms of physical pubescent changes, consistent with O’Neill et al. where these mothers embraced and supported their adolescent children with ID during their physical pubescent changes [43]. This is in contrast to mothers’ attitudes to changes in adolescent sexual behavior, where they expressed a range of views [43]. However, mothers did not address puberty or sexuality topics beyond menstruation, self-care, and the Janaba bath, which is mainly influenced by Islamic teaching. Menstruation seems to be one element of puberty that mothers felt confident explaining to their daughters since they had learned it formally and informally in Islamic education in the mosque or at school [44]. On the other hand, when there is a hint of nocturnal emission, mothers taught their sons in an understandable way about the Janaba bath.

They also stated that SE should be age-appropriate and tailored to the individual’s mental development, and education should be at a level that is understandable by children, similar to previous studies [38,45,46]. To meet parents’ perceived requirements and unmet needs of children with ID, Sravanti and Jayendra offered a model that described the “What, Why, and How” of five constructs to ease in providing SE: understanding human body, hygiene and self-care, social skills, privacy, safety, and handling feelings [46].

However, educating children with disabilities is very challenging. Almost all educated mothers said that they need to educate children based on mental capabilities and not based on their chronological age. That is why these mothers used every opportunity they had to discuss the topic of sexuality indirectly with their children rather than educating them directly. Previous studies found that parents prefer to wait for their children to initiate the conversation about sex, or even until the children already have a particular sex-related issue, due to the difficulties in initiating the topic [47,48]. Some of these mothers in our research also took advantage of every opportunity to give their children ongoing SE. When a family member was pregnant, they discussed how the baby grows and is delivered from a woman’s body. Additionally, they advised and taught their children indirectly, especially when they engaged in improper sexual behaviors.

#### 4.3.2. The Less Educated Mothers

Our findings clearly indicated that children’s disability, socio-cultural norms, and other environmental factors have a significant effect on less-educated mothers’ decisions to offer SE to their children. For these mothers, the thought of discussing sexuality generates trepidation, and depending on the children’s level of comprehension, this may lead to the selection of a specific topic for discussion. Our findings are consistent with prior works of literature, which indicated that parents are concerned that SE may exacerbate their children’s sexual desires and result in improper sexual behavior [37,39]. Despite the debate surrounding SE, research indicates that it does not promote sexual activity but rather promotes abstinence and equips adolescents with the information and skills essential for appropriate sexual behavior [49]. Children with ID may have cognitive and behavioral capacities that are substantially delayed in terms of their physical development and sexual desires. This discordance may result in improper sexual practices and societal distress. SE should be comprehensive and incorporate all aspects of sexuality. It includes providing knowledge on human sexual anatomy and reproduction, sexual intimacy, sexual pleasure, sexual orientation, gender identity, sexual restraint, contraceptive methods, sexually transmitted diseases, and reproductive rights and responsibilities [5].

We also found that the less-educated mothers in our study used more figurative language as compared with educated mothers to refer to reproductive organs instead of using the appropriate terminology, as is customary in Malay. Our findings contrast with previous research conducted among Malaysian parents in which the vast majority of parents (93.8%) reported educating their children on the correct names for their genital organs [25]. At a young age, parents should begin teaching their children age-appropriate terminology for body parts and their functions as early as 4 years old [25]. There is mounting evidence that using anatomically precise terms such as penis, scrotum, vagina, and vulva benefits children’s early development of body confidence, self-esteem, and safety [50,51]. Many parents are better at teaching their children to recognize their anatomy with general, amusing, or diverting terms. There is a considerable worry that euphemism conveys the notion that these parts are shameful, humiliating, or secretive nature. By referring to these parts by their accurate terminology, discussions on maintaining them in healthy, clean, and safe ways can be facilitated. Parents should speak clearly and without fear of stigma or embarrassment. Using the right terminology and speaking freely about these subjects may prepare the children to voice future worries about sexual health, exploitation, or abuse and, most crucially, it may ensure children’s safety [51].

All mothers in our study were Malay Muslims. Thus, when they taught their children, they incorporated religious beliefs as a component of SE. They taught their children the concept of halal and haram in Islam and the concept of shyness. Shyness serves as an inner constraint, allowing Muslims to sustain proper behavior and morals [47]. One study conducted among Indonesian parents to explore their perceptions on providing SE to children with ID found that incorporating religious rules in sexual education made them feel better and made it easier to communicate [39]. This research is also consistent with the study by Elias et al., in which they conducted their qualitative study among Muslim mothers in Pasir Puteh, Kelantan. According to mothers, SE is already ingrained in their everyday lives as Muslims, and from an Islamic perspective, teaching SE must begin with shyness [52].

We acknowledge the study’s shortcomings. Despite attempts to engage mothers and fathers from many ethnic backgrounds, the endeavor was in vain. It represented, however, the actual ethnic composition of Kelantan, where 94% of the population is Malay and the majority of children’s primary caregivers are women. Future research should involve parents from various sociodemographic backgrounds or geographical regions. Furthermore, a mixed-method approach may be used to obtain comprehensive results on parents’ knowledge, attitudes, and practices about SE of their children with disabilities.

## 5. Conclusions

Some mothers of children with ID had poor understandings regarding sexuality, and they viewed sexuality primarily through the lens of sexual encounters and intimate relationships. Mothers’ views on sexual behavior varied, as the majority of mothers did not foresee their offspring engaging in sexual behaviors. Sexually explicit events were often considered inappropriate or disagreeable for children with ID. Sexuality and SE can be an extremely challenging topic of the discussion between mothers and their children with ID. Therefore, mothers practice different approaches when providing SE to their children, either following physicality or socio-cultural methods. This study emphasizes the need for strategies to increase parents’ understanding and awareness regarding sexuality and appropriate delivery of SE to their children with ID, and that it should be a shared responsibility between healthcare providers, teachers, and social workers to help manage their children’s emerging sexuality effectively.

## Figures and Tables

**Figure 1 ijerph-19-07249-f001:**
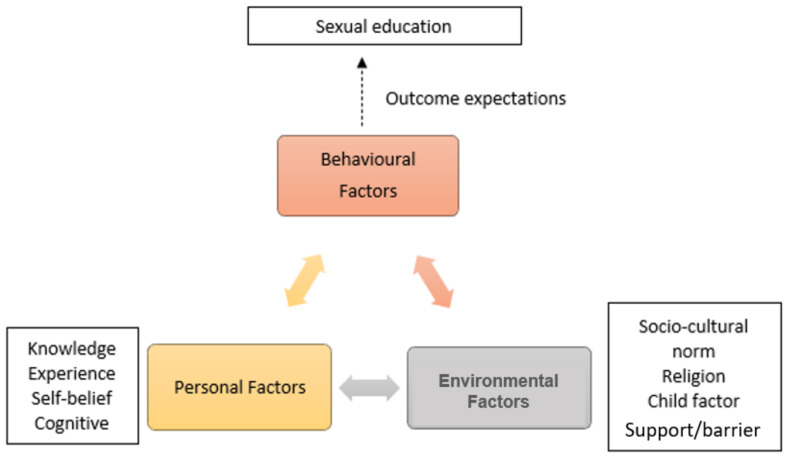
Bidirectional factors that influence mothers’ outcomes based on Social Cognitive Theory [26].

**Figure 2 ijerph-19-07249-f002:**
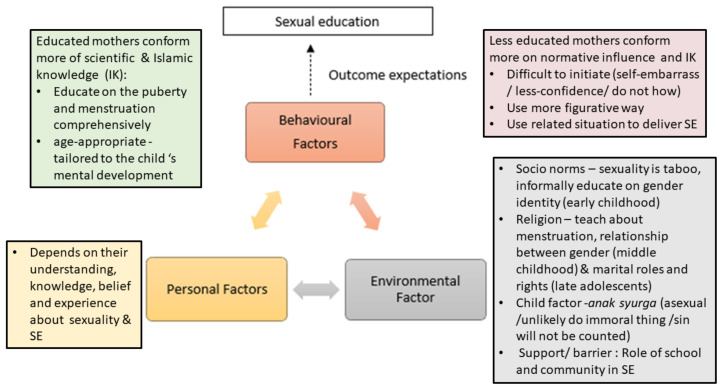
Effect of personal factors, sociocultural norms, and behavior factors on mothers’ choices regarding sexual education practices.

**Table 1 ijerph-19-07249-t001:** Semi-structured questionnaire: an interview guide.

Introduction		
This study concerns sexuality and sexual education (SE) for children with intellectual disabilities (ID). I am interested to hear your experience and your perspective as parents in describing sexuality and providing SE to your children living with ID and how environmental factors influence your SE practice. Before we go further, would you please tell me about yourself? Your children? Your family? Your job?		
Questions	Specific Questions	Probing Questions
From your experiences, what does sexuality mean?	What does sexuality mean to you? How do you get personal messages about sexuality?	Describe how environmental factors (upbringing, school, and community) influence personal messages about sexuality. How does this understanding influence your view and SE practice on your child?
	How do you view sexuality in your children as a sexual being?	Describe in terms of sexual development, behavior and sexual maturity? How do your children (with ID) convey or express sexuality (in any way)? A way of dressing, gender identity, a way of thinking, behavior and attitude. Any differences with your other children? If yes, how?
2. What are your experiences in providing SE to your child with ID?	What are your views on SE in Malaysia?	What do you know about SE? How important is SE to children with ID? If it is, why? If not, why? What is your most concern in delivering SE? Any difference in providing SE among your children (with and without ID)? In what condition will SE help your children now and in the future?
	How confident are you to convey SE to your children?	How early did you initiate the conversation? Any specific topic? Any difficulty in communicating SE with your child with ID? If yes, why. How much did your child with ID understand the topic discussed? Please describe.
Ending: Before ending this interview, is there anything else that you want to add, comment, or share? Asking their hopes, recommendations, advice or sharing words about this issue with other parents with ID children. Thank you.		

**Table 2 ijerph-19-07249-t002:** Characteristics of the mothers and children with intellectual disabilities (n = 20).

Variables		Mean (SD)	n	%
**Mother’s Age (years)**		48.75 (8.45)		
**Marital Status**				
	Married		14	70
	Widowed/Divorced		6	30
**Mother’s Educational Level**				
	Primary (Standard 1 (7 years old)—Standard 6 (12 years old))		3	15
	Secondary (Form 1 (13 years old)—Form 5/6 (17/18 years old))		11	55
	Tertiary (Completed college/university)		6	30
**Mother’s Occupation**				
	Homemaker		11	55
	Public Sector		6	30
	Private Sector		3	15
**Mother’s Religion**				
	Muslim		20	100
**Child’s Age (years)**		15.35 (2.62)		
	Early Adolescent (10–14)		7	35
	Middle Adolescent (15–17)		7	35
	Late Adolescent (18–19)		6	30
**Child’s Sex**				
	Male		15	75
	Female		5	25
**Child’s Diagnosis**				
	Intellectual Disability solely		5	25
	Autism Spectrum Disorder with comorbid ID		4	20
	Down Syndrome		5	25
	Mixed with comorbid ID		6	30
**Number of Siblings**		4.45 (1.61)		
**Position in the Family**		2.95 (1.85)		
First			4	20
Second			6	30
Third and above			10	50

## Data Availability

Not applicable.

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
