# Peer review of "“Providing Sex Education Is Challenging”: Malay Mothers’ Experience in Implementing Sex Education to Their Children with Intellectual Disabilities"

_ijerph, 2022, doi:10.3390/ijerph19127249_

Round 1
Reviewer 1 Report
Using qualitative in-depth interviews with Malay mothers who have children with mild to moderate intellectual disabilities (ID), this study documented the meaning of sexuality as interpreted by the mothers and how sex education would be beneficial to adolescents with ID. Although this qualitative study is interesting, revisions are needed for publication in peer-reviewed journals
I believe I have reviewed a manuscript that was from the same study. As such, I will provide similar comments. First, to prepare the reader, I suggest that the authors develop a couple of paragraphs in the introduction section to describe social and cultural contexts in Malaysia that are relevant to sexuality and sex education. In particular, I would like to see a description of the religious and cultural norms that are relevant to sexuality and sex education and detailed information about the settings (e.g., schools, homes) and major government or non-governmental agencies that are responsible for sex education in Malaysia.
Second, it would be very important for readers to understand sexuality and sex education in terms of the Islamic doctrines and religious traditions, which can be used to highlight the need for sex education among children who are intellectually disabled.
Third, the theoretical framework was appropriate but misplaced. Instead of introducing the theoretical framework in the discussion section, I would develop a literature review section to follow the Introduction. In so doing, this theoretical framework can be used as a guide for data analysis and directions for future research.
Author Response
Dear reviewers,
Thank you for all the comments and suggestion. The team has been working on it to follow the suggestion for improvement.
Thanks again.

Reviewer 2 Report
Thanks for the opportunity to review this manuscript. I benefitted from reading this manuscript and hope to see it eventually published. I offer some suggestions to make the manuscript stronger for publication.
Introduction, para. 1: the following statement is unclear to me: “Sexuality involves the development of physical, mental, and social behaviors of every human being.” I’d recommend re-wording it a bit.
Introduction, para. 3: “Sexuality issues of persons with ID are still misjudged by society even though this group exhibits comparable normal sexual desires akin to the general population”. I’d suggest removing the word “normal” here.
Introduction, para. 7: “A limited study has been carried out in Malaysia on parents’ views on sexuality and sex education for their children with ID [17].” I was curious about why this study is limited and what it found. More importantly, I was curious to hear about how this present study contributes to address some of the existing gaps.
Introduction, para. 8: I appreciated the discussion about social learning theory but I believe that this paragraph can use a bit more unpacking. For example, what exactly are “personal factors”? What about environmental factors?
Introduction: My main feedback about this section is that the contribution of this paper is not clearly articulated. What does it add to the literature reviewed? What are the key contributions?
Methods, procedure: “The interviewer was a female Family Medicine Trainee with no prior relationship to the participants.” I was curious about the last part here. Could this also be a limitation? Perhaps mothers who may not be as comfortable sharing their experiences without some prior relationship? Just curious.
Themes: “Many mothers just do not comprehend or are unaware of what sexuality is.” This is unclear to me. Don’t they all have some interpretation of it nonetheless? Please clarify. This could use some unpacking. How many? I am not getting a clear sense in the findings.
Themes: The way the text is formatted it is a bit hard to separate the authors’ interpretation of the data and the actual quotes from the participants.
Themes: overall, this is a strong section and it packs a lot of data. I really appreciated the quotes shared.
Discussion: Here, again, I’d encourage the authors to make more links to the theory. It is mentioned in the beginning of this section and we have a visual diagram; however, I’d have liked to see more connections between made between the findings and the theory. In what ways does this study contribute to this theory or vice versa?
Conclusion: I believe this paper makes a contribution to the literature. I’d encourage the authors to make that contribution more clearly articulated here. That’d serve to highlight the importance of the study.
Minor comments:
Instead of using the term “able-bodied”, I’d use the term “non-disabled.” This is a just a way to decenter able-bodiedness/mindness.
I’d add a couple more keywords, one about the methods another related to the location of the study perhaps.
Author Response
Thanks for all comments and suggestions. We have amended as suggested.
Regards,
RDM

Reviewer 3 Report
The authors report on Malaysian mothers and their perceptions to sex education for their children with intellectual disabilities. Although I think that the paper is not evolved sufficiently yet to be published, I compliment them with their efforts to address this important issue. Below I’ll give my major and minor concerns, hopefully these help the authors to bring their paper to a next level.
Major:
- Authors state (line 134-136): “We began by asking the participants the following key questions: “From your experience, what does sexuality mean?”; “What is your view about sexuality in your children?” and “How do you provide sex education to your children?” These were then following by more specific questions.” These are correct open questions, however, I wonder how the authors directed to children with ID? I assume that mothers had more children, also without ID. So, how did authors distinguished children with and without ID?
- Authors have used qualitative analyses to sort their data. However, the results read rather anecdotical and elaborate on specific cases in length. Although these observation are interesting, I miss the big picture. Authors are encouraged to deepen their analyses to improve their paper. In addition – but related – in perspective of the study aims, it only makes sense to report on mothers’ own perceptions of sexuality if they relate these findings to how these mothers perceive and practice sex education to their child with ID. The same holds for the child’s sexual development.
- Authors used the SCT as theoretical framework. However, this model should be explained in the introduction, and incorporated in the methods section as how this model shaped the interview protocol. Then, as mentioned above, the SCT should direct the analyses. As a result, in my opinion this would imply the necessity to rewrite the discussion.
Minor:
- General advice to involve a native speaker to improve the writing
- Line 42: please write ID first in full, and use ID consequently in paper (i.e., not learning disability, IDs)
- I suggest to rearrange the introduction: move the 5th alinea lines 65-74, to the 2nd alinea (start from line 42), and then place the 4th alinea
- Line 113: please define mild to moderate ID
- Line 123: how did the researchers determine whether saturation was reached?
- In the results section, please present all percentages in rounded numbers without decimal (e.g., 70% instead 70.00% because N=20
- In the results section, please put quotes in quotation marks and italics
- In the results, references to participant identification (i.e., P.10) does not add to the results because the reader does not know her background. Consider to delete
- Table 1: the mothers’ and childs’ ages are expressed in means. I am not sure what the authors mean by ‘(month)’. I presume they mean ‘year’?
- Table 1: please specify young/middle/late adolescence (instead of adolescent) in methods
- Table 1: please specify ‘Mixed with comorbid ID in methods
- Table 1: please specify ‘position in the family’ in methods. Present this as categorical results (i.e., first, second, etc) instead of mean
- Line 188: please reformulate. It reads as a value judgement instead of result
- Line 217: please reformulate ‘limited view’, it reads as value judgement. Idem line 241.
- Line 326: what does ‘akil baligh (puberty)’ mean. This is explained later on, but it should be clear from the start
Author Response
Dear reviewers,
Thanks for all comments and suggestion for improvement given to us. We have tried our best to accommodate all the comments and amended accordingly.
Thanks again
RDM

Round 2
Reviewer 1 Report
I am satisfied with the revisions. Though the study cannot be generalized to other settings in Malaysia, the study findings are essential for studies on children who have an intellectual disability. I enjoyed reading the paper.
Author Response
Dear reviewer,
Thanks for the good remark. Attached is the amendment for reviewer 3

Reviewer 2 Report
I am satisfied with the changes made. Thanks.
Author Response
Dear reviewer,
Thanks for the good remark. Attached is the amendment for reviewer 3. Thanks.

Reviewer 3 Report
The authors have improved the manuscript significantly. My first major point is addressed in the rebuttal, but I suggest to also address this in the manuscript. (Authors state: “We began by asking the participants the following key questions: “From your experience, what does sexuality mean?”; “What is your view about sexuality in your children?” and “How do you provide sex education to your children?” These were then following by more specific questions.” These are correct open questions, however, I wonder how the authors directed to children with ID? I assume that mothers had more children, also without ID. So, how did authors distinguished children with and without ID?)
Authors incorporated the SCT model in the introduction. However, I miss its implication in the methods (how did the SCT model shape their interview guide) and in the results (how do the results fit in the SCT model). The results still read anecdotal which in my opinion needs more in depth analyses. For example, how is it justified to see “Sexuality is an intimate relationship” and “Sexual intimacy” as two distinguished subthemes? In my view, structuring the results according to the SCT model would be helpful. Note: the three themes occur in all three SCT components.
Minor:
- Please update the abstract after rewriting the manuscript
- Title and citation information do not match
- Line 157: Social Cognitive Theory instead of social learning theory
- Add reference to title figure 1
- Explain in Methods how Educational levels are defined
- Use round percentages in the text (30% instead of 30.0%)
- Table 1: widowed instead of widow
-
Author Response
Dear reviewer,
Thank you for your comments of improvement. The amendments are as below.
